# 2D Convolutional Neural Markov Models for Spatiotemporal Sequence Forecasting

**DOI:** 10.3390/s20154195

**Published:** 2020-07-28

**Authors:** Calvin Janitra Halim, Kazuhiko Kawamoto

**Affiliations:** 1Department of Applied and Cognitive Informatics, Graduate School of Science and Engineering, Chiba University, Chiba-shi, Chiba 263-8522, Japan; calvinjh@chiba-u.jp; 2Graduate School of Engineering, Chiba University, Chiba-shi, Chiba 263-8522, Japan

**Keywords:** spatiotemporal forecasting, time series prediction, deep neural networks, deep Markov model, CNN, LSTM, DMM

## Abstract

Recent approaches to time series forecasting, especially forecasting spatiotemporal sequences, have leveraged the approximation power of deep neural networks to model the complexity of such sequences, specifically approaches that are based on recurrent neural networks. Still, as spatiotemporal sequences that arise in the real world are noisy and chaotic, modeling approaches that utilize probabilistic temporal models, such as deep Markov models (DMMs), are favorable because of their ability to model uncertainty, increasing their robustness to noise. However, approaches based on DMMs do not maintain the spatial characteristics of spatiotemporal sequences, with most of the approaches converting the observed input into 1D data halfway through the model. To solve this, we propose a model that retains the spatial aspect of the target sequence with a DMM that consists of 2D convolutional neural networks. We then show the robustness of our method to data with large variance compared with naive forecast, vanilla DMM, and convolutional long short-term memory (LSTM) using synthetic data, even outperforming the DNN models over a longer forecast period. We also point out the limitations of our model when forecasting real-world precipitation data and the possible future work that can be done to address these limitations, along with additional future research potential.

## 1. Introduction

Time series forecasting has long been a challenging problem in computer science. Depending on the sequence to be modeled, parameters might have either 2D or 3D spatial dependencies, which often arise in real-world phenomena, e.g., weather, sea movements, and other similarly physically governed phenomena. Most of these spatiotemporal sequences cannot be easily measured and predicted accurately because of the inherent noise in the dynamics and measuring equipment. A class of forecasting methods known as data assimilation was invented specifically to solve this problem, which models the target sequence using probabilistic models such as Gaussian state-space models. Using Bayesian inference and measured observations, data assimilation methods forecast how the sequence evolves with reduced noise. The Kalman filter [1] and its derivatives, such as extended Kalman filter, unscented Kalman filter [2], and ensemble Kalman filter [3], are some well-known examples of data assimilation methods.

Data assimilation methods typically require that the sequences being modeled are based on a known physical equation, as they use numerical methods to model the evolution of the sequence. This modeling severely limits the application of the methods for sequences with unknown dynamics. Meanwhile, deep neural networks (DNNs), specifically, recurrent neural networks (RNNs), have also been used for sequence predictions because of their capacity to approximate the underlying dynamics of a sequence, even without knowing the parameters. Among several types of RNNs, long short-term memory (LSTM) [4] was conceived to address the shortcoming of vanilla RNN and has been the backbone of every modern DNN time series prediction method. One particular method to predict spatiotemporal sequences derived directly from LSTM is convolutional LSTM (ConvLSTM) [5]. By substituting every LSTM matrix operator with a convolutional neural network (CNN), the model leverages the spatial information encoded on the sequences and reduces the amount of memory required by the network.

However, DNN-based models are not without limitations when used to forecast spatiotemporal data. For example, ConvLSTM is ultimately a deterministic model that does not take into account the stochastic character of inherent system noise and observation noise of the target sequence. This might result in a higher forecast error rate in data with higher noise variance. Conversely, a combination of a probabilistic state-space model and DNN-based approach to model time series data, called the deep Markov model (DMM) [6], is a promising approach to model stochastic dynamics because of its structural similarity to data assimilation methods. Nonetheless, the original model is restricted to 1D data, making it challenging to capture the spatial characteristics of spatiotemporal data.

To address these problems, we propose a DMM that forecasts a spatiotemporal sequence, inspired by the spatial modeling structure of ConvLSTM, replacing every standard fully connected layer in the model with a 2D CNN layer. By doing so, we aim to leverage the spatial characteristics of a target sequence while still modeling the dynamics and observation noise of the sequence. To summarize, we propose a novel methodology for spatiotemporal forecasting with the following contributions:The method introduces a DMM that maintain the spatial structure of the input data by running them through a full 2D model, which consists of several 2D CNNs and a backward ConvLSTM, with the intention of capturing the inherent spatial features of the data. Using DMM as a base model allows the integration of probabilistic modeling to spatiotemporal forecasting problem, increasing the robustness of the proposed approach.The feasibility of our method is evaluated by conducting two experiments using a synthetic spatiotemporal data modeled after 2D heat diffusion equation, as well as real-world precipitation data. We compare the results with other baseline models, namely, naive forecast, DMM, and ConvLSTM.The combination of 2D CNNs, ConvLSTM, and DMM in the proposed approach opens up the possibility of combining popular 2D CNN-based methods, further increasing DMM’s modeling capability to cater to various spatiotemporal forecasting problems. Conversely, the proposed approach also allows the usage of DMM in other fields, such as video prediction and generation, due to its autoencoder-like structure.

The rest of the paper is organized as follows. Section 2 presents several related approaches in spatiotemporal forecasting, focusing on ConvLSTM and DMM-based models, along with their relation to our method. Section 3 shows the formulation of the spatiotemporal task in general. Section 4 explains the detailed training flow, prediction flow, and the model structure of our method. Section 5 presents the experiment details and results, showing the feasibility and limitations of our method when compared to other baseline models. Section 6 focuses on the result of the experiments, challenges that arise from it, together with future research to address the challenges and to improve upon the current method. Lastly, Section 7 summarizes the content and findings of the paper.

## 2. Related Work

Research regarding spatiotemporal forecasting using DNN-based methods has advanced very rapidly in recent years. Le et al. [7] transformed air pollution data into a series of images that were then fed into ConvLSTM to forecast future data. Elsayed et al. [8] modified the internal gate structure of ConvLSTM to reduce the parameters required. Even though not strictly a physical spatiotemporal problem, [9] combined ConvLSTM with seq2seq framework and stochastic RNN to forecast financial trading data, presenting an alternative application of the model. On the topic of the combined approach of CNN and LSTM other than ConvLSTM, [10] combines CNN encoder with the autoencoder version of LSTM to forecast electricity data, managing to achieve best performance when compared to other DNN methods. The favorable performance of the autoencoder structure in [10] supports the motivation for autoencoder-like structure in our method. Meanwhile, approaches such as [11] and [12] use a specific class of model called a graph neural network (GNN) to model traffic data, as traffic data is more suitable to be modeled using GNN instead of CNN because of its non-Euclidean structure.

The use of a DMM to infer and model 2D input data is not new [13,14,15]. The general approach taken by these models is utilizing layers of 2D CNNs and flattening the output of the last layer to encode 2D data into 1D data that can be processed by DMM. The same approach is taken to reconstruct the input data, with CNNs converted into deconvolutional neural networks (DCNs). This differs from our approach, as the 2D structure of the data is lost halfway through the models.

There is one particular model that also combines the convolutional paradigm with a DMM [16], similar to ours. However, their approach fundamentally differs from ours, as they use a temporal CNN [17] instead of a spatial CNN, with the goal of modeling speech features for use in recognition and classification tasks. To our knowledge, our approach is the first that uses a 2D CNN to retain the 2D spatial structure of the input sequence throughout the DMM.

As the task of video prediction in a broad sense can also be thought of as a spatiotemporal forecasting problem (2D sequence with an additional dimension of color channel), there have also been advancements in this field of research that we note can also be applied to general spatiotemporal forecasting tasks. For example, [18] combined a CNN variational autoencoder and adversarial training regime to produce multiple "realistic" and possible future frames. Another example sees [19] utilizing a version of CycleGAN [20] with a generator trained to be bidirectionally (past and future) consistent and two discriminators to identify fake frames and sequences. Nevertheless, we note that most research in this field is geared toward achieving a qualitatively "realistic" result, which differs from our goal of accurately modeling the evolution of a sequence. The process of producing diverse predictions in these models, however, might be able to be adapted to spatiotemporal forecasting research, in general, to produce several possible outcomes of a sequence, increasing understanding of the target sequence dynamics. We leave this approach for possible future work.

## 3. Spatiotemporal Sequence Forecasting Task

The task of spatiotemporal forecasting can be defined as follows. First, we define the 2D gridded spatial observation of an event with *M* rows, *N* columns, and *K* measurements as a tensor $X\in {I\;R}^{K\times M\times N}$. Note that even though the acquired observation data is spatially a 2D matrix, there could be multiple measurements taken within the same space; hence, the 3D tensor definition is appropriate. A 2D spatial observation can then be observed along the temporal dimension, and we define observations that are observed within the first and T timesteps (inclusive) as $X_{1:T}$.

When given an input of past observations $X_{1:T}$, we can thus define forecasting as the task of calculating an estimate of an unobserved future sequence from timepoint T+1 to T+ΔT, written as ${\hat{X}}_{T+1:T+\Delta T}$, in which the sequence has the highest probability to occur. ΔT here stands for the time difference between the start and the end of the forecasted sequence. As described by [5], this task can be defined by the following equation.
(1)$$\begin{array}{c} {\hat{X}}_{T+1:T+\Delta T}=\underset{X_{T+1:T+\Delta T}}{arg max}p\left(X_{T+1:T+\Delta T}|X_{1:T}\right) \end{array}$$

Depending on the task description, the forecasting task might not be defined as picking the most plausible sequence but rather generating a set of highly plausible sequences. As our approach is based on a DMM, which is a stochastic model, our approach will generally produce a sample of the set of possible predictions, i.e., ${\hat{X}}_{T+1:T+\Delta T}∼p\left(X_{T+1:T+\Delta T}|X_{1:T}\right)$. However, when trained correctly, we note that our approach will produce forecast samples with high probability, and in the comparison, we regard our model as generating predictions with the highest probability.

## 4. 2D Convolutional Neural Markov Model

### 4.1. Overview and Structure of the Model

Our approach is mostly based on the DMM, consisting of two separate models, an inference network, and a Gaussian state-space model-based generative network, trained using the variational inference method. Utilizing variational inference and state-space model, the model can learn to approximate the plausible state-space model that governs the sequences, which in effect is similar to data assimilation. Indeed, [6] and [21] have shown that the DMM can match the estimation capability of the unscented Kalman filter [2], which supports our motivation to use the DMM as the base of our method.

We describe the general flow of the model when it is used to forecast an observation sequence. Given a 2D observation sequence $X_{1:T}$, we want to first infer the posterior latent $Z_{1:T}$ that gives rise to the observation. Following the variational inference paradigm of [6], we can use the inference network to infer an approximation of the true posterior probability $p(Z_{1:T}|X_{1:T})$, denoted by $q_{\phi }\left(Z_{1:T}|X_{1:T}\right)$. ϕ here denotes the network parameters of the inference network. Afterward, we sample the last posterior latent of the sequence ${\hat{Z}}_{T}$ from the approximated latents and use it as the initial input for the generator network. We then propagate through the generator network from the input to produce the next latent ${\hat{Z}}_{T+1}$ and then the forecasted observation ${\hat{X}}_{T+1}$ of the next time point. These procedures can be repeated to produce a forecast of the required length.

To accommodate 2D spatiotemporal sequences, instead of reshaping them into 1D sequence data, we modify both the inference and generator networks to accept 2D data by changing all of the matrix multiplication operators on the model into 2D CNNs. By changing these operators, we are able to reduce the size of the operators and the models, compared with fully connected ones, which directly reduces the redundancy and the tendency of the model to overfit. Furthermore, using CNNs, we can capture not only the temporal characteristics but also the spatial information encoded on the data in a hierarchical mean. More importantly, this also opens up the door to apply existing CNN techniques and research into the DMM, reinforcing the representation capability of the model.

### 4.2. Inference Network

The inference network we use is the same as the structured inference network derived by [6], in which the posterior latent sequence is factorized as follows:(2)$$\begin{array}{c} p\left(Z_{1:T}|X_{1:T}\right)=p\left(Z_{1}|X_{1:T}\right)\prod_{t=2}^{T}p\left(Z_{t}|Z_{t-1},X_{t:T}\right) \end{array}$$
along with the similarly structured approximated posterior (approximated using a Gaussian distribution):(3)$$\begin{array}{c} q_{\phi }\left(Z_{1:T}|X_{1:T}\right)=q_{\phi }\left(Z_{1}|X_{1:T}\right)\prod_{t=2}^{T}q_{\phi }\left(Z_{t}|Z_{t-1},X_{t:T}\right) \end{array}$$
where ϕ denotes the set of inference network model parameters. In [6], the inference network models the Markovian structure of the approximated posterior with a combination of backward LSTM and a combiner network. The LSTM will propagate the input sequence in reverse, i.e., from the future (according to Equation (Equation 3)), outputting a series of hidden outputs that will then, along with the previous latent, be used as inputs for the combiner network. The combiner network will produce the estimated latent.

However, we note that our input is 2D spatiotemporal data, and we also want to preserve the spatial structure throughout the model. Thus, we introduce several modifications to the network. First, we encode the input sequence using layers of the CNN encoder layer, which reduces the spatial size of the input to reduce the parameter size of the network. The encoded input will be fed into a backward ConvLSTM, preserving the 2D structure. The hidden tensors from ConvLSTM will be propagated to a combiner CNN, along with the latent tensor of the previous timestep, to produce the latent tensor of the current timestep. The combiner CNN follows the structure of the combiner network defined in [6], while changing the matrix multiplication operators into 2D CNNs.

The flow of this network is graphically shown in Figure 1 and can be described by the following equations, when given an input sequence $X_{1:T}$:(4)$$\begin{array}{cc} {\tilde{X}}_{1:T} & =\mathrm{Encoder}(X_{1:T}) \end{array}$$(5)$$\begin{array}{cc} H_{t},C_{t} & =\left\{\begin{array}{cc} \mathrm{ConvLSTM}({\tilde{X}}_{t},H_{t+1},C_{t+1}), & t<T\\ \mathrm{ConvLSTM}({\tilde{X}}_{t},H_{init},C_{init}), & t=T \end{array}\right. \end{array}$$(6)$$\begin{array}{cc} {\hat{Z}}_{t} & =\left\{\begin{array}{cc} \mathrm{Combiner}({\hat{Z}}_{t-1},H_{t}), & t>1\\ \mathrm{Combiner}({\hat{Z}}_{0},H_{t}), & t=1 \end{array}\right. \end{array}$$
where Encoder is the encoder layer, consisting of 2 layers of a 2D CNN with ReLU activations:(7)$$\begin{array}{cc} {\tilde{X}}_{t} & =\mathrm{Encoder}\left(X_{t}\right)=\mathrm{ReLU}({\mathrm{CNN}}_{X^{\prime }}\left(\mathrm{ReLU}\left({\mathrm{CNN}}_{X}\left(X_{t}\right)\right)\right) \end{array}$$

The CNNs used here reduce the spatial size by half (stride=2,kernel=(3×3),padding=1), resulting in a total of 1/4 th reduction. ConvLSTM is a ConvLSTM cell [5]. Combiner is the combiner function, defined by:(8)$$\begin{array}{cc} H_{combined} & =0.5(\tanh\left({\mathrm{CNN}}_{H}\left({\hat{Z}}_{t-1}\right)\right)+H_{t}) \end{array}$$(9)$$\begin{array}{cc} {\hat{Z}}_{\mu ,t} & ={\mathrm{CNN}}_{{\hat{Z}}_{\mu }}\left(H_{combined}\right) \end{array}$$(10)$$\begin{array}{cc} {\hat{Z}}_{\sigma ,t} & =\mathrm{softplus}({\mathrm{CNN}}_{{\hat{Z}}_{\sigma }}\left(H_{combined}\right)) \end{array}$$
where ${\mathrm{CNN}}_{H}$, ${\mathrm{CNN}}_{{\hat{Z}}_{\mu }}$, and ${\mathrm{CNN}}_{{\hat{Z}}_{\sigma }}$ are 2D CNN layers. Note that the CNN layers used in ConvLSTM and the combiner function are size-preserving CNNs, i.e., CNNs with stride=1, kernel=(3×3), and padding=1. Here, we also note that $H_{init}$, $C_{init}$, and ${\hat{Z}}_{0}$ are all trainable parameters (tensors) that are initialized to zero tensors.

Using Combiner, the posterior latent for the current timepoint *t* can then be sampled as follows:(11)$$\begin{array}{c} {\hat{Z}}_{t}∼N\left({\hat{Z}}_{\mu ,t},{\hat{Z}}_{\sigma ,t}\right) \end{array}$$

The sampled posterior can then be used to calculate the posterior for the next timepoint, and this process is repeated until the whole posterior sequence is calculated. The posterior mean and variance calculated here are in the form of 2D matrices with an additional channel dimension, assuming that the variables are independent of each other. We tried modifying the model to accommodate multivariate dependence, but because of the enormous weight parameters required for the covariance calculation, the model quickly becomes intractable. This also applies to the sampling procedures of *Z* and *X* in a generative network. We leave this research regarding an alternative approach to integrating multivariate dependence for future work.

As mentioned earlier, the inference network we build here uses the temporally backward version of ConvLSTM, following the derived factorization shown in Equation (Equation 3). Even though several other approaches (mean-field, forward, and bidirectional factorizations) are evaluated in [6], the results show that backward factorization yields a model with sufficient modeling capability. Therefore, in this paper, we focus our experiment on models with backward factorization.

### 4.3. Generative Network

The fundamental structure of the generative network is based on a Gaussian state-space model [6], consisting of a transitional function to propagate the latent sequence and an emission function to calculate the corresponding observation of each latent. We directly mimic the structure given by [6], replacing each neural network operator with 2D CNN layers but maintaining the activation functions.

Concretely, for a transition function, we replace the fully connected networks with a gated transition function described in [6] with 2D CNNs:(12)$$\begin{array}{cc} G_{t} & =\mathrm{sigmoid}({\mathrm{CNN}}_{G^{\prime }}\left(\mathrm{ReLU}\left({\mathrm{CNN}}_{G}\left(Z_{t-1}\right)\right)\right)) \end{array}$$(13)$$\begin{array}{cc} J_{t} & ={\mathrm{CNN}}_{J^{\prime }}\left(\mathrm{ReLU}\left({\mathrm{CNN}}_{J}\left(Z_{t-1}\right)\right)\right) \end{array}$$(14)$$\begin{array}{cc} Z_{\mu ,t} & =\left(1-G_{t}\right)⊙{\mathrm{CNN}}_{Z_{\mu }}\left(Z_{t-1}\right)+G_{t}⊙J_{t} \end{array}$$(15)$$\begin{array}{cc} Z_{\sigma ,t} & =\mathrm{softplus}({\mathrm{CNN}}_{Z_{\sigma }}\left(\mathrm{ReLU}\left(J_{t}\right)\right)) \end{array}$$

Similar to the combiner function in the inference network, the CNNs defined here are all size-preserving CNNs. Propagation of the latent sequence starts at $Z_{0}$, which is a learnable parameter initialized as the zero tensor during training. Sampling of the next latent is done as follows, using the calculated mean and variance:(16)$$\begin{array}{c} Z_{t}∼N\left(Z_{\mu ,t},Z_{\sigma ,t}\right) \end{array}$$

This will then be used to propagate the latent matrices to the next timestep, a process that is repeated until the end of the sequence.

To produce the corresponding reconstructed observations, we adopt a 1-layer size-preserving CNN and 2-layer DCNs (stride=2,kernel=(4×4),padding=1) with ReLU activation functions, following the structure of the emitter function described by [6]:(17)$$\begin{array}{cc} L_{t} & =\mathrm{ReLU}({\mathrm{DCN}}_{L}\left(\mathrm{ReLU}\left({\mathrm{CNN}}_{L}\left(Z_{t}\right)\right)\right)) \end{array}$$(18)$$\begin{array}{cc} X_{\mu ,t} & =\tanh({\mathrm{DCN}}_{X_{\mu }}\left(L_{t}\right)) \end{array}$$(19)$$\begin{array}{cc} X_{\sigma ,t} & =\mathrm{softplus}({\mathrm{DCN}}_{X_{\sigma }}\left(L_{t}\right)) \end{array}$$

The use of a DCN instead of CNN is to increase the size back to the original size, which in this case will increase the size by 4 times (2 times increase with each DCN), matching the reduction of the encoder layer in the inference network. The observation can then be reconstructed by sampling $X_{t}∼N\left(X_{\mu ,t},X_{\sigma ,t}\right)$ throughout the sequence. The generative network is graphically described in Figure 2.

### 4.4. Training and Forecasting Flow

#### 4.4.1. Training Procedure

The training flow is shown by Figure 3a. Following [6], during training, *n*-set of $K_{X}\times M_{X}\times N_{X}$ (channels×height×width) observation sequences $\left\{X_{1,1:T},X_{2,1:T},\ldots ,X_{n,1:T}\right\}$ is fed into the inference network to infer a set of $K_{Z}\times M_{Z}\times N_{Z}$ posterior latents $\left\{{\hat{Z}}_{1,1:T},{\hat{Z}}_{2,1:T},\ldots ,{\hat{Z}}_{n,1:T}\right\}$. The approximated latents will be used to reconstruct the observation sequences using the generative network and to estimate the conditional likelihood $p(X_{1:T}|Z_{1:T})$ along with KL divergence. These will be used to calculate the factorized evidence lower bound (ELBO) as an objective function for each sequence: (20)$$\begin{array}{c} L\left(X_{1:T};\left(\theta ,\phi \right)\right)=\underset{q_{\phi }\left(Z_{1:T}|X_{1:T}\right)}{E}\left[\log p_{\theta }\left(X_{1:T}|Z_{1:T}\right)\right]\\ -\mathrm{KL}\left(q_{\phi }\left(Z_{1:T}|X_{1:T}\right)\left|\right|p_{\theta }\left(Z_{1:T}|X_{1:T}\right)\right) \end{array}$$
where θ is the parameters of the generative network and ϕ is the parameters of the inference network. The model is then backpropagated and updated using gradient descent-based algorithms. Note that while [6] derived the analytic solution to ELBO, in this paper, the models are trained using Monte Carlo gradient estimation. In addition, we use the Adam optimizer to train the model.

#### 4.4.2. Forecasting Flow

The procedure we use to forecast a sequence is described as follows. When we are given a sequence of observations $X_{1:T}$, in which we want to forecast ${\tilde{X}}_{T+1:T+\Delta T}$, we first feed the past observations $X_{1:T}$ into the inference network to acquire the posterior latents ${\hat{Z}}_{1:T}$. Afterward, we input ${\hat{Z}}_{T}$ into the transition function of the generative network to output the predicted latent ${\tilde{Z}}_{T+1}$. This predicted latent will then be fed into the generative network’s emission function, outputting the forecasted observation ${\tilde{X}}_{T+1}$. At this point, we can then continue our forecast in two different ways:**Multi-step method**: By repeating these generative steps recursively, we can produce a forecast sequence with an arbitrary length, i.e., repeat the steps ΔT−1 times to output ${\tilde{X}}_{T+1:T+\Delta T}$. This method requires a very well-trained generative network to be accurate, as problems such as high variance or biased calculation produced by suboptimally trained transition and emitter functions will result in chaotic predictions.**One-step method**: Instead of forecasting every observation point with only the generative network, we instead update our observations in real-time when we have new ones, and time-shift the input to the inference network by 1 ($X_{2:T+1}$), acquiring new posterior latents ${\hat{Z}}_{2:T+1}$. We use the newly estimated ${\hat{Z}}_{T+1}$ to estimate ${\tilde{Z}}_{T+2}$, and in turn ${\tilde{X}}_{T+2}$. Finally, we then repeat this procedure to produce the rest of the forecast. Note the similarities of this method to data assimilation, in which we keep updating our estimates using newly obtained observations. This forecasting method is shown in Figure 3b.

In the following experiment, we focus on evaluating our model’s forecasting capability using the one-step method.

## 5. Experiments and Results

### 5.1. 2D Heat Equation

In the experiments, we aim to evaluate the model forecasting capability and its stability with respect to forecasting noisy data. To do this, we generate a Gaussian state-space model toy problem from a 2D heat equation, essentially simulating a randomly positioned circle of heat on a 10 m × 10 m gridded plate that dissipates over time. We define the heat equation as follows: (21)$$\begin{array}{cc} \frac{\partial U}{\partial t} & =D\left(\frac{\partial^{2}U}{\partial x^{2}}+\frac{\partial^{2}U}{\partial y^{2}}\right), \end{array}$$(22)$$\begin{array}{cc} \mathrm{Transition} & :Z_{t}∼N\left(\mathrm{FD}\left(Z_{t-1}\right),3I\right), \end{array}$$(23)$$\begin{array}{cc} \mathrm{Emission}1 & :X_{t}∼N\left(Z_{t},I\right), \end{array}$$(24)$$\begin{array}{cc} \mathrm{Emission}2 & :X_{t}∼N\left(Z_{t},10I\right) \end{array}$$

*D* here expresses the thermal diffusivity (set as 4.0$m^{2}/s$), (x,y) is the location on the grid, *U* expresses the temperature, and *I* is the identity matrix. Note that there is only one parameter (temperature), making the size of the input channel 1; hence, the matrix form can be used instead of the tensor. The initial temperature for the circle of heat is randomized between 500 K and 700 K, and the radius of the circle is randomized between 0.5 m and 5 m, with the central position of the circle randomized within the range of the plate. Meanwhile, the base temperature is initialized as 0 K.

We then use the finite difference (FD) method to calculate the temperature evolution, with the addition of Gaussian noise, as shown in Equation (Equation 21). We also prepare two emission functions to produce the observations, which are both Gaussian noise with a ten times increase in variance in the second one, to depict an increase in noisiness. The spatial differences dx and dy used in the FD method are set to 0.1 m (producing a sequence of heatmap data with the size of 100×100 pixels), and the timestep difference is set at 0.000625 s. We generate 3000 simulations for training data by first generating 100 simulations and taking 30 sequential samples randomly from the simulations with a length of 30 timesteps and time difference three times of 0.000625 s. This is done to mimic how data is measured in the real world, evaluating robustness to noisiness and chaos. We also generate 750 simulations as validation data with the same method. We summarize the details in Table 1.

### 5.2. CPC Merged Analysis of Precipitation

For evaluation of real-world data, we opt to utilize the CPC Merged Analysis of Precipitation (CMAP) data provided by the NOAA/OAR/ESRL PSL, Boulder, Colorado, USA, from their website at https://psl.noaa.gov/ [22]. These data show the global gridded precipitation rate measured from rain gauges, combined with estimates from satellite-based numerical algorithms. The detail of the merging is described in [22]. There are two versions of the data available: the first one is data with monthly values from 1979-01 to 2020-05, while the second one is pentad-valued data from 1979-01 to 2017-01 (as of 2020-06-28). In this paper, we choose the pentad-valued data for our evaluation, as there is a smaller time difference between the data and a larger dataset (497 timesteps for monthly data, compared to 2774 timesteps for pentad data). We use the enhanced version of the data, which combines the satellite and rain gauge estimation with blended NCEP/NCAR Reanalysis Precipitation values to fill out missing estimations and measurements. The blended precipitation values are forecasted values whose method is described in [23].

The data consist of a 2D gridded sequence, totaling 2774 timesteps. We divided the sequence into an overlapping sequence of 30 timesteps, with a ratio of roughly 7:3 for training and validation data, yielding 1901 sequences for training data and 815 for validation data. Note that we divide the data so that no overlap occurs between the training data and validation data. As the spatial size of the data is not a square (72×144 pixels), we crop the center of the data into square data with a size of 72×72 pixels for ease of evaluation. Like the 2D heat equation data, there is only one parameter being measured, resulting in an input channel of 1. The details of the data’s attributes are summarized by Table 2.

### 5.3. Model Specification and Experiment Details

For the experiment, our model’s specification follows the details described in Section 4. However, we vary the channel sizes of CNNs and DCNs in our model for each experiment, as shown in Table 3.

For the comparison baselines, we use a naive forecast method that regards the observation of the previous timestep as current forecast, a 1-layer ConvLSTM [5] and a vanilla DMM [6]. For ConvLSTM, the hidden channel is 64 and the kernel size is 3×3; forecasting is performed using the previous 10 hidden matrices, concatenating them on the channel dimension, and applying a 1×1 convolution layer, then retrieving the prediction for the next timestep. We do not vary the parameter for ConvLSTM between experiments.

Regarding the vanilla DMM, to accommodate the 2D spatial data, we add a 2-layer CNN encoder layer before the LSTM layer, just like our model, with a channel size of (32, 64), and then add a fully connected network, producing a 256-parameter 1D sequence. This sequence is run through a 128-parameter backward LSTM layer, which will, in turn, be fed into a combiner layer that produces a 50-parameter latent variable. The generative network consists of a 64-parameter gated transitional function and a 2-layer DCN emitter, which is also similar to our model, with the same channel size (64, 32). The kernel size, stride, and padding used here are the same as in our model when applicable (including the downsampling and upsampling process in the encoder and decoder respectively), and they do not vary between experiments.

To prevent the KL divergence term from overpowering the objective loss during initial training epochs for our model and the DMM, we employ an annealing factor during training with a minimum value of 0.2 and increase it linearly for each epoch. We utilize the Adam optimizer to train all models with the parameters shown in Table 4. Normalization of data values by scaling them into values within the range of −1 and 1 is performed before inputting the data into the model, by setting a data range of 0 to 1000 K for 2D heat diffusion data and a data range of 0 to 80 mm/day for CMAP data. Here and in the experimental results, our model is shown as the convolutional neural Markov model (CNMM). All of the models are implemented in PyTorch and Pyro [24]. The codes for the experiment can be found at GitHub.

For both experiments, observation mean squared error (MSE) with respect to ground truth is used as the evaluation metric. Instead of calculating the error of the sampled observation, we calculate the mean produced by the Emitter (for ConvLSTM, we use the forecasted observation directly, and in the case of the naive forecast, we took the MSE of the ground truth with the 1-timestep shifted version of it). Here, we evaluate our approach (CNMM), ConvLSTM, vanilla DMM and naive forecast method on the first and second conditions of the heat experiment (**Emission 1** and **2**, as shown in Equations () and ()) and CMAP data with varying forecasting length (5, 10, 15, and 20 timesteps). When evaluating with the **Emission 2** condition, we utilize the models trained on the first emission. This is to evaluate the robustness of the models with respect to noise. To ensure fairness, forecasting for every model uses the one-step forecast method. Furthermore, we run the training procedure five times (except naive forecast, as there are no training required) and present the averaged MSE from every last epoch of the run as the final result. We present the resulting forecast MSE in both table and bar chart forms. The forecast MSE for 2D heat equation data is shown by Table 5 and Figure 4, while the CMAP data is shown by Table 6 and Figure 5. Note that the MSE is calculated on normalized data instead of unnormalized data.

Other than the MSE of the forecasts, we also plot the spatially averaged squared error of nine random samples taken from the validation data, which can be seen in Figure 6 and Figure 7 for 2D heat equation data and CMAP data respectively, to aid understanding of the experimental results. Additionally, we show the squared error heatmap between the ground truth and forecasted values, also for both 2D heat equation and CMAP data in Figure 8 and Figure 9. Specifically, we plot the squared difference of the forecasts from the first sample (sample used by the upper left plot) shown in Figure 6 and Figure 7. Finally, we present the heatmap visualization of forecast results on the same validation sample, shown by Figure 10 and Figure 11. We only show the graphs from Emission 1 samples to represent 2D heat equation forecast result, instead of showing both Emission 1 and 2 samples as Emission 2’s forecast result shows a similar trend. In addition, the forecast result given by naive forecast is not directly shown in the heatmap visualization as it can easily be inferred by shifting the ground truth heatmap forward by one timestep.

### 5.4. Experimental Results

#### 5.4.1. 2D Heat Equation

As shown in Table 5 and Figure 4, we first notice that results given by naive forecast surpasses every result produced by DMM-based models. Examining Figure 6 and Figure 8 reveals that the naive forecast error is mostly either comparable or lower than DMM-based models, hence the superior results. We deduce that this shows that the dynamics generated by the heat equation and emission conditions are low enough to be modeled by the naive forecast. This also shows that DMM-based models are inferior when used to model dynamics with comparatively low variance. Nevertheless, we conducted further experiments with noisier emission conditions, and the results show that DMM-based models, including ours, can achieve lower forecast errors compared to naive forecasts. This is also confirmed by the lower errors seen in the CMAP data, which will be investigated further in the next section.

Focusing solely on the deep learning-based models, Table 5 and Figure 4 show that our approach managed to surpass the accuracy of ConvLSTM on longer prediction length and surpass the modeling capability of vanilla DMM in both emission conditions. We do note that even though Figure 6 shows that there are some samples where vanilla DMM yields comparable forecast error compared to our model, our model yields a less noisier forecast error, which explains the robustness of our model compared to vanilla DMM. We can also see that the error increase in our model is relatively small compared with that in DMM when the forecast length is increased (20 timesteps). This shows the stability of our forecasting method over more extended periods, even when presented with noisy data. On the other side, Figure 6 and Figure 8 shows that ConvLSTM suffers from a comparatively huge error in the first 2 timesteps of every sample given, explaining the lower error forecast compared to DMM and our model. We believe that this is due to the combination of the usage of the initial hidden unit, which is set to zero (as it does not have the proper hidden unit before initial forecast), and the steep initial descent of temperature as exhibited by some of the data. This might also show that our model is more robust to sudden changes in the data compared to ConvLSTM. We plan to investigate more regarding this behavior in the future.

Ultimately, however, for shorter prediction lengths, ConvLSTM yields better predictions. Looking at the result shown by Figure 6, it is indicated that ConvLSTM yields a very low error of prediction after a few timesteps, compared to DMM, our model, and even naive forecast. This shows that ConvLSTM, in general, has a superior modeling ability compared to variational and naive methods. Even though the ConvLSTM initially outputted huge error, it quickly returns to a more stable forecasting error along with time. The reason this can happen is that the ConvLSTM prediction result is smoother than that of ours and the DMM because of its nonstochastic nature, as can be seen in the heatmap visualization given by Figure 10.

Still, we note that our model outperforms the forecasting capability of the DMM in both conditions. This shows that our model, which incorporates a 2D spatial structure throughout the model, manages to take advantage of the structure to infer the underlying dynamics of the data more accurately than that using the 1D structure of the DMM. We also conducted a short experiment using the multi-step method to forecast the heat equation on Emission 1. However, our model performs poorly compared with ConvLSTM, falling into a chaotic state even after three steps of prediction. We deduce that this is due to the variational structure of the model, in which the trained generative model could have learned a comparatively huge variance, leading to an exploding error when propagated.

#### 5.4.2. CMAP

Table 6 and Figure 5 shows that when evaluated using CMAP data, our model underperformed for all forecasting lengths when compared with both the DMM and ConvLSTM, with ConvLSTM yielding a slightly better result compared with the DMM. This is supported by further examination on Figure 7, suggesting that our model produced a slightly higher error compared to DMM and ConvLSTM. We hypothesize that our model suffers from suboptimal training because it has comparatively many layers of CNN, making it prone to problems such as vanishing gradient and overfitting. In fact, when we first use the parameters we set to train 2D heat equation data (larger channel sizes) on CMAP, we found out that the model overfits after approximately 10-15 epochs of training and yields a worse result compared with the case when we decrease the parameters to those presented in Table 6.

We also note that the CMAP data basically has less training data compared with our synthetically generated heat equation data, as there are many overlaps of the data in each training sequence, compared with the nonoverlapping sequences in our data (as we can generate new data easily with the underlying dynamics). A lower number of data points results in a biased and overfitted model, which our model is especially prone to because of the large number of parameters.

Moreover, as can be seen by the results, the variational inference-based models (DMM and ours) yield worse results compared with ConvLSTM. This might also be caused by the difficulty in training the variational model, primarily because we used Monte Carlo estimation to calculate the gradients. The stochasticity of the variational models might also have played a part, as we employ the Gaussian state-space model to model the dynamics of the data, the effectiveness of which should increase with a more extensive training dataset.

Even with all of the problems and limitations presented above, we confirm that our model, along with vanilla DMM and ConvLSTM, are able to surpass the naive forecast method. This proves that the model is still usable as a better forecasting method than naive forecast when the target dynamics of the data are chaotic enough.

## 6. Discussions

Our evaluation of the synthetic heat equation data proves that there is an advantage to incorporating 2D CNNs inside the DMM, with our model outperforming the DMM for all forecast lengths and conditions, and also outperforming ConvLSTM when the forecast length is increased and when there is a steep change of values in the data. However, the evaluation using the real-world data shows the limitation of our model compared with the baseline models, with weaker accuracy and the tendency to overfit when given a smaller training dataset. Nevertheless, our model still yields better forecasts than that of naive forecast in a chaotic real-world setting, demonstrating that our model could be considered as one of the alternative approaches to model real-world data.

Elaborating on the training of our model, we hypothesize that a different structure or configuration of our model might yield a better prediction compared to baseline models even in a less noisy environment. As an example, before changing our model into a bottleneck configuration on the encoder-decoder CNNs (downsampling and upsampling structure), we also tried training our model with strictly size-preserving CNNs. The results show that although training (convergence) is much faster, the forecast accuracy presented was ultimately suboptimal compared with our model’s bottlenecked version. We then tried various hyperparameters and channel sizes to increase the training and modeling capability of our model, and we managed to gain a slight increase in the accuracy, even though still not enough to surpass the baseline models. We think that a more rigorous evaluation of the effect of hyperparameters is required, which we are presently researching.

Another configuration that we tried was applying batch normalization in specific layers of the CNN (encoder, decoder, and combiner functions) to help regularize the layers. Applying batch normalization makes the training harder, as expected, but it did not increase the accuracy of our model, even though it is plausible that another configuration might yield a better result. Indeed, this shows that different configurations of the model will yield a different result, demonstrating the diversity of our model.

Furthermore, we also have to note that during the training of our model, there is a chance that the model will train suboptimally compared to other trial runs, yielding a worse performance. We regard these runs as outliers in the experiments. As mentioned above, we believe this is due to the high number of parameters combined with the probabilistic aspect of the model. Preliminary experiments show that given the right hyperparameters (a bigger batch number paired with a bigger channel size), a more stable model can indeed be acquired, albeit with a precondition of more extensive and comparatively unbiased data.

Even with its limitations, our approach paves the way for assimilating the diverse research concerning CNNs into the DMM, with models such as residual networks (ResNets) [25] as a possible solution to the difficulty in training (by solving the vanishing gradient problem). Our work also shows that given sufficient compatibility, modifying the DMM with other related DNN models is also a promising area of research. One other alternative that can be explored is incorporating graph-based DNNs into the DMM. Because of its 2D spatial characteristics and unsupervised nature, as mentioned in Section 2, application of our model to video prediction and data generation is another possibility that can be studied in the future.

## 7. Conclusions

We proposed a model that combines the spatial structure of ConvLSTM and the variational technique of the DMM as an alternative method of spatiotemporal forecasting. Our evaluation shows that while there are some limitations and difficulty in forecasting data with a limited number of training data points and smaller variance, our model either matches or outperforms (in the longer forecasting period) other baseline models when utilized to forecast a stochastic system. In the future, we plan to perform more rigorous experiments with model configurations and research methods to improve our model’s forecasting capability.

## Figures and Tables

**Figure 1 sensors-20-04195-f001:**
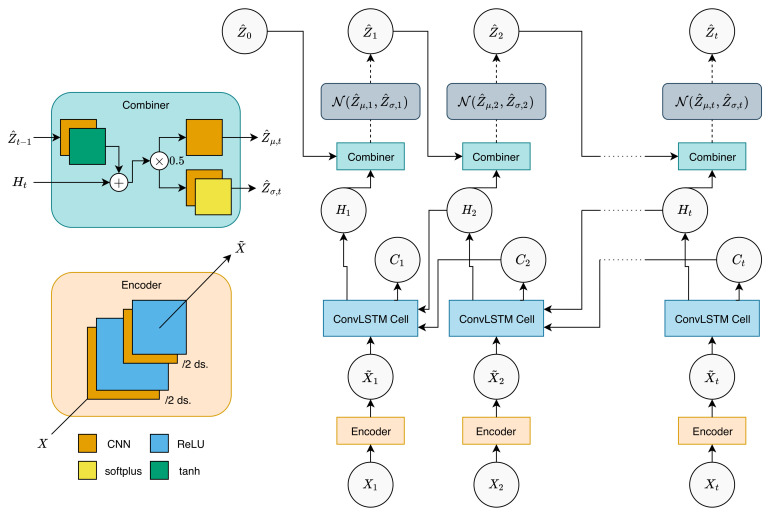
Inference network. Observation is first encoded using encoder to produce encoded observation. The encoded observation for a time point and hidden tensors from future timepoints are then fed into a backward ConvLSTM cell to produce hidden tensors for the current timepoint. When hidden tensors for a particular sequence are calculated, they are then fed into combiner along with the previous posterior latent to produce the current latent mean and variance. We then sample the current latent from the produced mean and variance, which follows a Gaussian distribution. Here, “/2 ds.” denotes 1/2 reduction in spatial size (downsampling). Dashed lines denote sampling, while dotted lines denote repetition. The description of the ConvLSTM cell structure can be found in [5].

**Figure 2 sensors-20-04195-f002:**
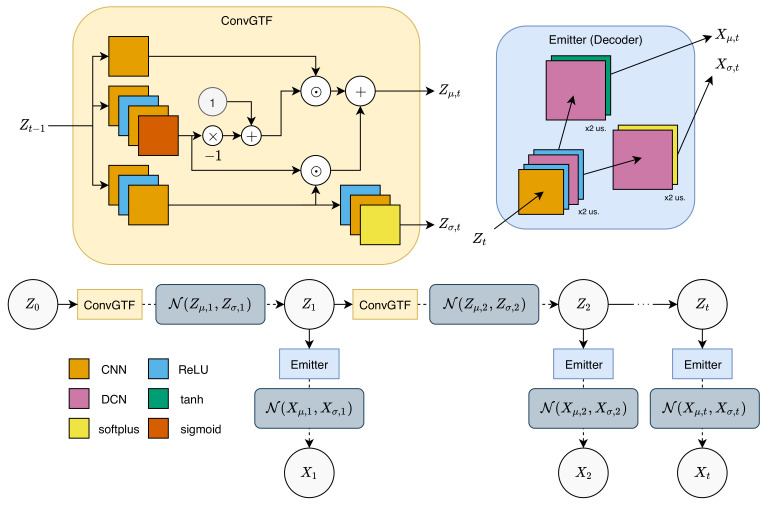
Generative network. Propagation on latent tensors is done by inserting the previous latent to a convolutional gated transition function (ConvGTF) to obtain latent mean and variance, which will then be used to sample the next latent. Observation is produced by inputting the latent into Emitter to output the observation mean and variance, which will be used to sample the observation. “x2 us.” denotes a two-times increase in spatial size (upsampling). Similar to Figure 1, dashed lines denote sampling, while dotted lines denote repetition.

**Figure 3 sensors-20-04195-f003:**
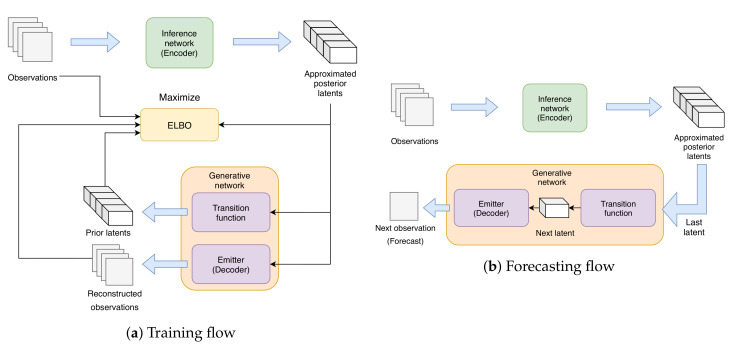
(**a**) shows the training flow of our approach. The observations are first fed into the inference network to produce a series of approximated posterior latents. These latents are then inputted into transition function to produce a shifted series of prior latents. The approximated latents are also inputted into emitter to produce reconstructed observations. Finally, the original and reconstructed observations, along with approximated posterior and prior latents are used to calculate the evidence lower bound (ELBO) as the objective function. (**b**) shows the one-step forecasting flow of our method. Similar to training flow, the observations are first inputted into inference network to produce posterior latents. We then use the last latent data to produce the next latent using transition function and calculate the next forecast using the emitter. We repeat this flow as new observations are obtained until the desired forecast length is reached.

**Figure 4 sensors-20-04195-f004:**
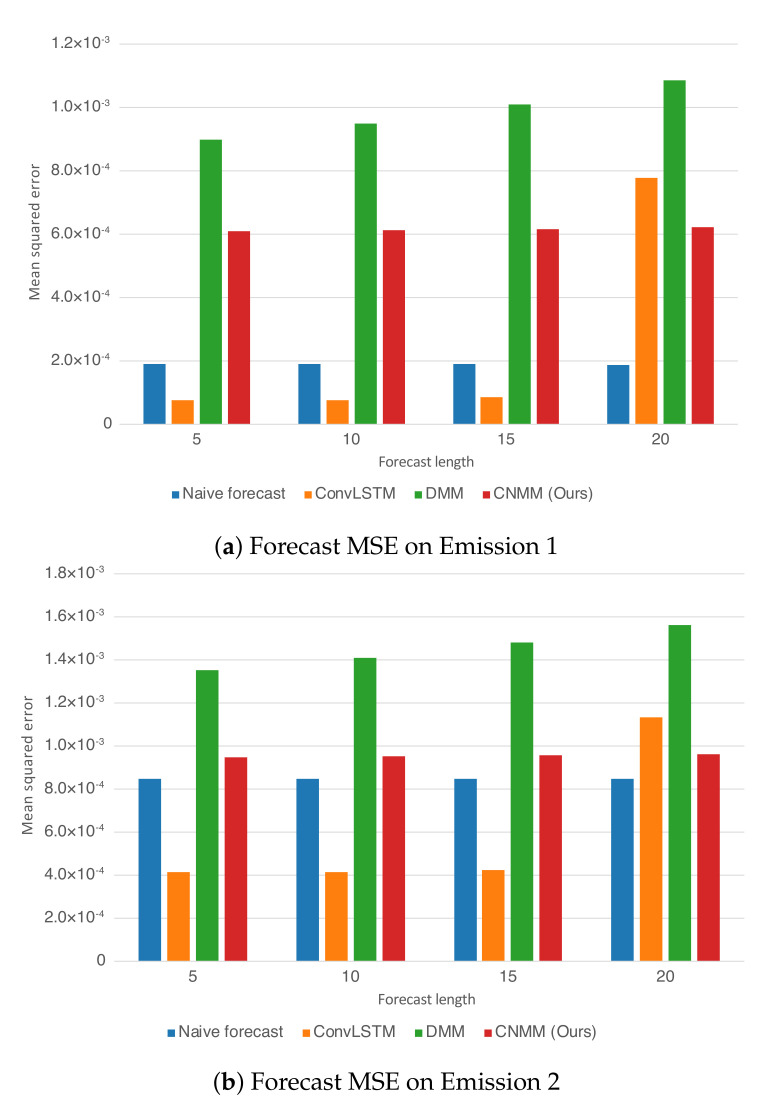
Forecast MSE on the CMAP data when plotted as bar graphs. The y-axis represents the MSE. The x-axis represents the forecast length and the model with which the forecast is produced

**Figure 5 sensors-20-04195-f005:**
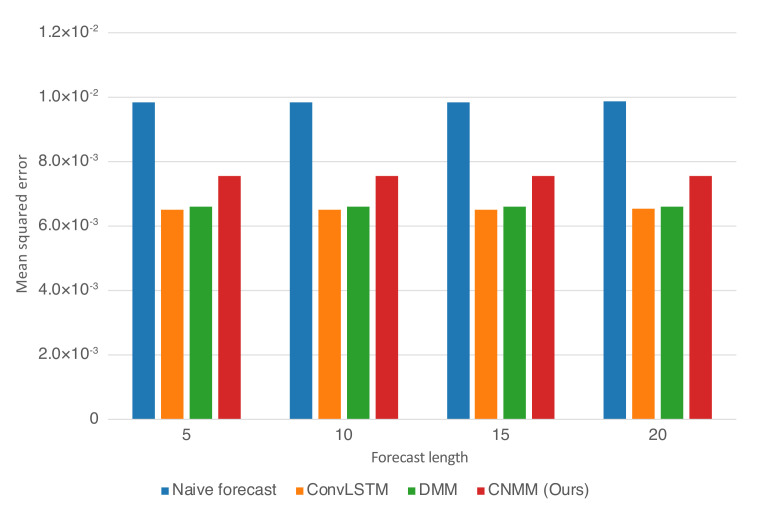
Forecast MSE on the CMAP data when plotted as bar graphs. The y-axis represents the MSE. The x-axis represents the forecast length and the model with which the forecast is produced.

**Figure 6 sensors-20-04195-f006:**
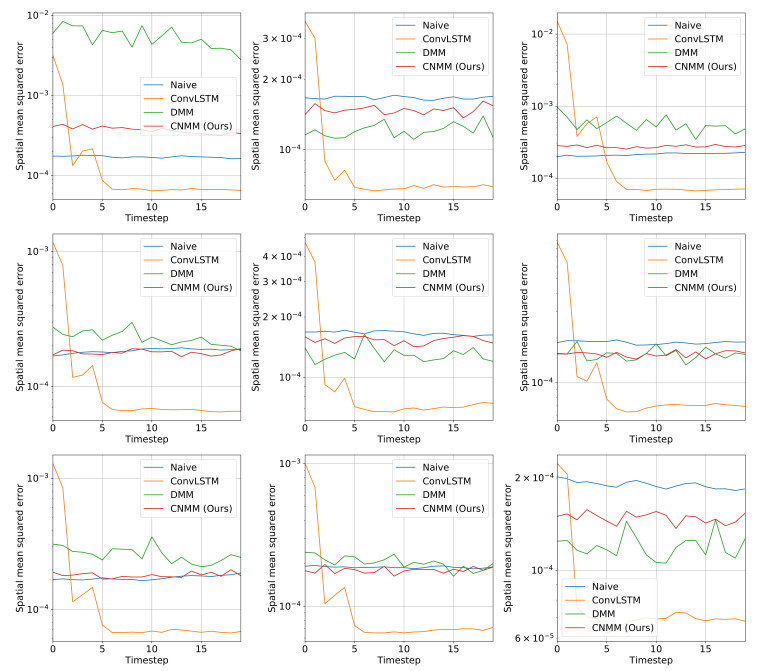
Spatially averaged (spatial mean) squared forecast error of baseline and our models on nine randomly selected 2D heat equation validation data. The data used here is from the Emission 1 condition, and the length of the forecast is 20. The y-axis shows the error value, and the x-axis shows the timestep of the forecast. It is clear that ConvLSTM’s forecasts underperformed on the first initial steps, while other models are more stable at forecasting the dynamics of the data.

**Figure 7 sensors-20-04195-f007:**
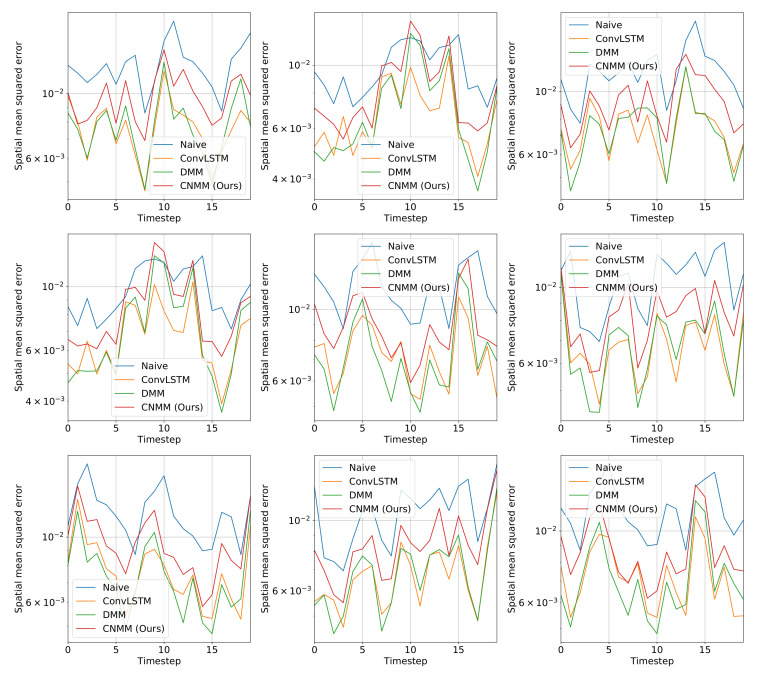
Spatially averaged (spatial mean) squared forecast error of baseline and our models on nine randomly selected CMAP validation data. The length of the forecast is set as 20 timesteps. The y-axis shows the error value, and the x-axis shows the timestep of the forecast. Compared to 2D heat equation data, every DNN model outperformed the naive forecast as expected, due to the higher variance and chaos introduced in the real-world data that are better modeled by DNN models.

**Figure 8 sensors-20-04195-f008:**
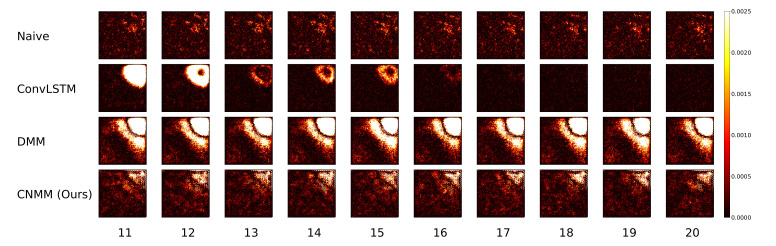
Squared error heatmap between ground truth and predicted forecast of the upper-left data used in the plot shown in Figure 6 (2D heat equation data). Higher errors are shown by the bright regions. Here, we can see the initial high prediction error in the area of heat on ConvLSTM’s forecasts, correlating with the high prediction error shown by Figure 6.

**Figure 9 sensors-20-04195-f009:**
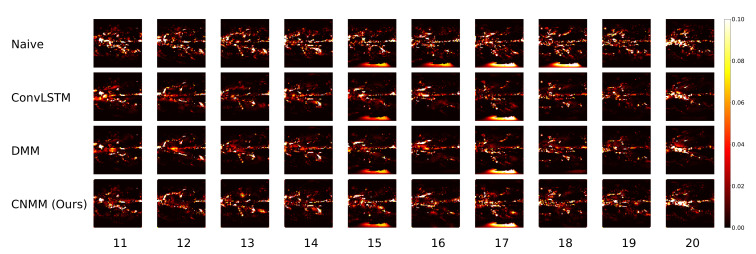
Squared error heatmap between ground truth and predicted forecast of the upper-left data used in the plot shown in Figure 7 (CMAP data). A brighter color means higher error.

**Figure 10 sensors-20-04195-f010:**
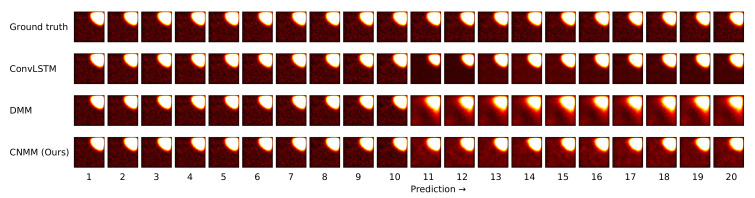
2D heatmap visualization of the forecast result of the same data used in Figure 8 (2D heat equation data). Prediction starts from the 11th timestep onward. Brighter region means higher temperature.

**Figure 11 sensors-20-04195-f011:**
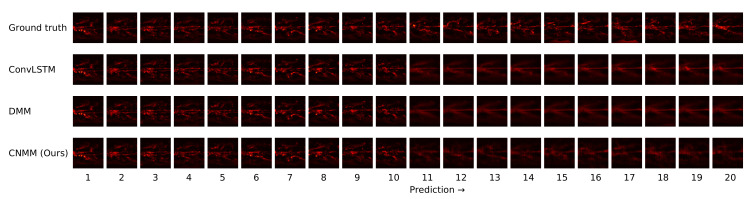
2D heatmap visualization of the forecast result of the same data used by Figure 9. As with Figure 10, prediction starts from 11th timestep and brighter region means higher precipitation level.

**Table 1 sensors-20-04195-t001:** Generation details of the 2D heat equation data. The random(min,max) here means that the parameter is sampled from a uniform distribution with specified minimum (min) and maximum (max) values.

Attributes	Value
(Minimum, maximum) values	(0, 1000) K
Plate size (width×length)	10×10 m
Thermal diffusivity	4.0 $m^{2}/s$
Base temperature	0 K
Initial temperature of circle of heat	random(500,700) K
Radius of circle of heat	random(0.5,5) m
Central position of circle of heat ((x,y))	(random(10,10),random(10,10)) m
Differentiation method	Finite difference
Spatial differences ((dx,dy))	(0.1,0.1) m
Timestep difference	0.000625 (×3 from sampling) s
Sequence length	30 timesteps
(Training, validation) data size	(3000, 750)

**Table 2 sensors-20-04195-t002:** Details of CPC Merged Analysis of Precipitation (CMAP) data.

Attributes	Value
(Minimum, maximum) values	(0, 80) mm/day
Spatial size (width×length)	72×72 pixels
Sequence length	30 timesteps
(Training, validation) data size	(1901, 815)

**Table 3 sensors-20-04195-t003:** Our model’s CNN and DCN channel specifications. () denotes multiple layers, while one value means the channel size is the same among all CNN inside the corresponding section. Heat denotes specifications during the 2D heat equation experiment and CMAP denotes specifications during the CMAP experiment.

Exp.	Inference			Generator	
	Encoder	ConvLSTM	Comb.	Trans.	Decoder *
Heat	(32, 64)	64	64	64	(32, 16)
CMAP	(32, 64)	16	16	16	(16, 8)

* Emitter.

**Table 4 sensors-20-04195-t004:** Training specifications. (a, b) denotes the parameters used in both experiments: a shows the parameter in the heat experiment, b shows the parameter in the CMAP experiment. One value for the parameter means the value is the same across experiments.

Parameters	ConvLSTM	DMM	CNMM
Learning rate (LR)	0.001	0.0001	(0.00005, 0.0001)
$\beta_{1}$	0.9	0.9	0.9
$\beta_{2}$	0.999	0.999	0.999
Grad. clipping	-	10.0	10.0
LR decay	-	1.0	1.0
Epoch	(100, 150)	150	(300, 150)
Batch size	16	16	16

**Table 5 sensors-20-04195-t005:** Forecast mean squared error (MSE) of the 2D heat equation data experiment on each model, with **Emission 1** and **Emission 2** denoting each emission condition of the data. The **length** here shows in how many timesteps the forecast is done on each model. The bold numbers indicate the lowest error achieved on each forecast length.

(**a**) Forecast MSE on Emission 1				
**Length**	**Heat equation - Emission 1** **(MSE** $\times 10^{4})$			
	**Naive Forecast**	**ConvLSTM**	**DMM**	**CNMM (Ours)**
**5**	1.8890	**0.7508**	8.9915	6.0846
**10**	1.8891	**0.7546**	9.5014	6.1196
**15**	1.8882	**0.8512**	10.0842	6.1591
**20**	**1.8876**	7.7907	10.8438	6.2114
(**b**) Forecast MSE on Emission 2				
**Length**	**Heat equation - Emission 2** **(MSE** $\times 10^{4})$			
	**Naive Forecast**	**ConvLSTM**	**DMM**	**CNMM (Ours)**
**5**	8.4880	**4.1452**	13.5044	9.4584
**10**	8.4820	**4.1485**	14.1001	9.5014
**15**	8.4751	**4.2451**	14.7963	9.5528
**20**	**8.4692**	11.3309	15.6192	9.6146

**Table 6 sensors-20-04195-t006:** Forecast MSE of the CMAP data experiment on each model. The **length** here shows in how many timesteps the forecast is done on each model. The bold numbers indicate the lowest error achieved on each forecast length.

Length	CMAP (MSE$\times 10^{3})$			
	Naive Forecast	ConvLSTM	DMM	CNMM (Ours)
**5**	9.8527	**6.5047**	6.6175	7.5480
**10**	9.8489	**6.5003**	6.6126	7.5447
**15**	9.8533	**6.5181**	6.6143	7.5501
**20**	9.8574	**6.5500**	6.6155	7.5529

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
