# Peer review of "2D Convolutional Neural Markov Models for Spatiotemporal Sequence Forecasting"

_sensors, 2020, doi:10.3390/s20154195_

Round 1
Reviewer 1 Report
The research is an interesting contribution to the emerging topic of spatiotemporal sequence forecasting. Literature review is good and up-to-date, the proposed architecture is new and potentially beneficial.
My main concern is related to demonstration of benefits of the proposed model specification (CNMM). The model architecture seems reasonable, but the experimental results show that a forecasting performance of CNMM is worse than of the regular ConvCNN. I very appreciate the Authors’ academic honesty, but an illustration of CNMM model advantages are highly required: is it really worth to use this model in practice and what are the pre-conditions for its efficient application? I would recommend to find the settings, where the proposed model will demonstrate its abilities to provide better forecasts.
Minor issues:
- Forecasting accuracy of simpler models would be beneficial for experimental data Tables 3 and 4. For examples, results from naïve forecasts and spatial averaging/krigging will provide a good benchmark for discussed models
- Please add the metric name (MSE) to the titles of Table 3 and 4 – this is quite difficult for the Reader to catch it from the paper text
- It would be better to use larger units (like MSE*10^3) for values in Tables 3,4: now this is difficult to compare the models due to many leading zeros
Reviewer 2 Report
The development of spatiotemporal forecasting models that do not rely on prior knowledge about parameters governing underlying system dynamics is at the forefront of the machine learning research. The authors propose an alternative method that accounts for stochastic dynamics and preserves the 2D structure of spatial data during model calibration and projections. Unfortunately, the demonstration of the proposed approach to two examples of “real-world” data indicates that the approach presented in this paper has “weaker accuracy and the tendency to overfit”. The validation analyses show that in most circumstances convolutional LSTM methods provide more accurate projections. Despite the limitations of the analysis the manuscript may be useful to guide the development of approaches to more accurately model spatiotemporal dynamics of chaotic and noisy systems.
Minor comments:
Would it be possible to show 2d figures of the modelled and estimated values for the applications with real-world data? It would be interesting to see how the authors’ approach approximates the spatial patterns of heat and precipitation.
Do you have an idea of how your approach could be applied to large scale 2D data? e.g. to analyse forest cover change at 1-hectare resolution at country scales.
L57 & L58 The sentences starting with reference numbers 7 and 8 look weird. Consider writing the authors names.
L59 Typo: parameter(s)
L60 Typo: representing -> represent
L 258 Are you planning to include a link to your GitHub repository here?
Tables 3 and 4. Indicate what is the meaning of the bold numbers and the numbers here. It is clear in the text but this information should be noted in the table.
Reviewer 3 Report
In this article, the authors proposed a model for sequence forecasting in which they combined the spatial structure of convolutional LSTM with vibrational approach of Deep Markov model. The presented idea is worthy therefore I recommend some minor suggestions to enhance the quality of the paper.
- The contributions are reflecting the novelty so it will be better to present in bullets in the last paragraph of the introduction section, for instance, you can see the paper https://www.mdpi.com/1424-8220/20/3/873/htm.
- In the current manuscript, the structure paragraph of paper is missing, therefore, the authors should mention it in the last paragraph of the introduction section.
- An important article on Autoencoder-LSTM is missing in the literature review, https://www.mdpi.com/1424-8220/20/5/1399/htm
- I did not find the visual representation of the proposed work; for better reader’s understanding, I suggest the authors to properly make the overall flow in the revised manuscript.
- In the experimental section, there is a lack of visual representation of the results achieved by proposed techniques such as graphs or tables. Furthermore, the authors should also provide the statistical details of the datasets used for the experiments.
Round 2
Reviewer 1 Report
The Authors carefully addressed all my comments.
Although the presented case studies are failed to clearly demonstrate benefits of the proposed model, I think this could be worth to publish the research at this state.
Reviewer 3 Report
The authors have addressed my comments well, I recommend it for publication now.